# Decoding Local Adaptation in the Exploited Native Marine Mussel *Mytilus chilensis*: Genomic Evidence from a Reciprocal Transplant Experiment

**DOI:** 10.3390/ijms26030931

**Published:** 2025-01-23

**Authors:** Marco Yévenes, Gonzalo Gajardo, Cristian Gallardo-Escárate

**Affiliations:** 1Laboratorio de Genética, Acuicultura & Biodiversidad, Departamento de Ciencias Biológicas y Biodiversidad, Universidad de Los Lagos, Osorno 5290000, Chile; ggajardo@ulagos.cl; 2Centro Interdisciplinario para la Investigación en Acuicultura, Universidad de Concepción, Concepción 4070386, Chile; crisgallardo@udec.cl

**Keywords:** functional genomics, differential gene expression, fitness-related outlier SNP-neighbor genes, blue mussels, conservation

## Abstract

Local adaptations are important in evolution as they drive population divergence and preserve standing genetic diversity essential for resilience under climate change and human impacts. Protecting locally adapted populations is essential for aquaculture species. However, high larval connectivity and frequent translocations challenge this in Chilean blue mussel (*Mytilus chilensis*) aquaculture, a world-class industry in Chiloé Island. This study examined local adaptations in two ecologically distinct natural beds, Cochamó (northernmost inner sea of Chiloé) and Yaldad (southernmost tip), through a 91-day reciprocal transplant experiment and genomic evidence. Cochamó mussels grew faster in their native environment (0.015 g/day) than Yaldad (0.004 g/day), though growth declined upon transplantation. Mussels transplanted within and between beds displayed distinctive adaptive transcriptomic responses, with differentially expressed genes involved with immune function, osmoregulation, metabolism, and cellular balance. Additionally, 58 known outlier SNPs mapped over the species’ genome sequence were linked with adaptive genes involved with osmoregulation, oxidative stress, and oxygen management, revealing selection-targeted specific genome regions. This study highlights how translocations affect the adaptive genomic response of *M. chilensis* and the impact of local environments in counterbalancing its genetic connectivity, concluding that the genomic differences in natural beds should be monitored and conserved for sustainable aquaculture practices.

## 1. Introduction

Local adaptations are fundamental in evolutionary and conservation biology, driving population divergence within species—the initial step towards speciation— and because they shed light on how natural selection shapes adaptive phenotypes and their genetic determinants in heterogeneous environments [1,2]. Populations that retain local adaptations and genetic diversity account for species resilience in a scenario of climate change and human-driven disturbances [3,4]. These adaptations persist over time as long as local selective pressures outweigh the homogenizing effect of gene flow [5,6,7]. Thus, understanding the genetic basis of adaptive traits in local populations is required for predicting their responses to environmental changes [8,9].

Besides climate change, human activities also threaten the adaptive potential of aquaculture species like marine mussels. Genetic homogenization can occur due to high larval dispersal and the translocation of juvenile mussels (seeds), which increase genetic connectivity. Seeds are often artificially collected from larval grounds close to natural mussel beds (seedbeds) and relocated to different grow-out sites, where they experience different selective pressures. Additionally, seed collection from seedbeds can impact the natural recruitment of these populations [10,11]. This scenario underscores the importance of applying evolutionary principles to the conservation and management of natural resources.

Conceptually, locally adapted populations exhibit higher average fitness in their native environment than any other population introduced [1], a phenomenon driven by genomic trade-offs across environments (e.g., antagonistic pleiotropy), meaning that no individual genotype excels across all habitats [2]. Reciprocal transplant experiments are commonly used as “acid tests” to assess local adaptation and apply two operational criteria: comparing the fitness of local individuals versus that of immigrants (local vs. foreign, δ_LF_) and assessing native fitness in the home versus non-native environments (home vs. away, δ_HA_) [12]. Reciprocal transplant experiments in oysters [13] and mussels [14,15] have demonstrated reduced survival, growth, and calcification rates and delayed reproduction in individuals transplanted to non-native environments, emphasizing the importance of considering local adaptations in aquaculture species, especially when individuals are transplanted outside their adaptive niche.

However, assessing fitness through reciprocal transplant experiments is often hindered by practical constraints, such as short-term or intragenerational studies, and biological challenges, such as long intergenerational time [1,2]. In such cases, exploring the underlying genetic foundations of fitness-related traits becomes useful, often achieved through genetic or genomic analyses [16,17]. At the genetic level, populations separated by geography or inhabiting distinct ecological niches are expected to exhibit unique genetic variants or differing frequencies of fitness–linked alleles in adaptive candidate genes [10,18]. Alternatively, genomic approaches have become standard methodologies, particularly those using next-generation sequencing (NGS) that detect fixed single-nucleotide polymorphisms (outlier SNPs) in genomic DNA and by analyzing differential gene expression through total RNA sequencing, bridging the gap between the genome and proteome [19]. While outlier SNPs located in genic (intronic, exonic) and regulatory regions stand out for their adaptive significance, providing essential information about the phenotype and the heritability of adaptive traits, total RNA sequencing (RNA-Seq) primarily targets whole-genome transcribed coding genes and, throughout differential expression analyses, have allowed for uncovering the genetic diversity of candidate genes that influence critical functional traits related to fitness and adaptive phenotypes in both model and non-model species [20,21,22,23,24,25]. In this context, the transcriptome, being a phenotype in itself [16], captures hereditary patterns of gene expression [26], integrating molecular and functional complexities throughout the genome [23,27]. It also provides a more accurate and realistic view of the plastic and adaptive genomic responses behind complex phenotypes, which are often controlled by multiple interacting genes [28,29,30]. Thus, since both play an essential role in local adaptation (differential gene expression and outlier SNP loci), these genomic approaches are particularly valuable when traditional fitness assessments via reciprocal transplant experiments are impractical.

*Mytilus chilensis* (Hupé 1854) is an endemic blue mussel of great significance in southern Chile due to its extensive use in aquaculture, making it an important biological model for investigating the role of habitat translocation (a standard industry practice), genetic variation, and differential gene expression in local adaptation. The species inhabiting rocky substrates in intertidal and subtidal zones along the South Pacific Ocean west coasts from Bío-Bío (38° S) to Magallanes (53° S) [31] has been subject to numerous ecological [32], eco-physiological [33], and adaptive genomic studies [34,35,36].

As a gonochoristic species, *M. chilensis* follows an annual gametogenic cycle, reaching sexual maturity in spring–summer. After fertilization, its planktonic larvae drift for 20 to 45 days before settling, potentially traveling up to 30 km depending on oceanographic conditions [37,38,39]. Such dispersal capacity has been used to explain weak genetic divergence and population structuring with neutral and other genetic markers. For instance, an allozyme analysis estimated an F_ST_ value of 0.03 [40], and microsatellites yielded an F_ST_ of 0.042 [41]. Another microsatellite-based study proposed that mussels from southern Chile represent a singular reproductive unit with no distinct regional stocks except for Punta Arenas in Magallanes [42]. At the genomic level and using neutral and outlier SNPs, some studies have investigated the genetic structure and evolutionary processes influencing marine populations of the species. Still, the interpretations of the results are controversial. For example, Segovia et al. (2024) [43], based on 5913 neutral SNPs and 50 outlier SNPs, concluded that genetic differentiation in *M. chilensis* is low (neutral SNP F_ST_: 0.004–0.015, outlier SNP F_ST_: 0.032–0.209), primarily shaped by neutral processes with weak signals of local adaptation, gene flow being the primary evolutionary force influencing the population genetic structure of the species. In contrast, Araneda et al. (2016) [44], using 891 neutral SNPs and 58 outlier SNPs, described genetic differentiation in adaptive loci (neutral SNP F_ST_: 0.005, outlier SNP F_ST_: 0.088–0.228) suggesting signs of local adaptation. Although the F_ST_ ranges for neutral and outlier SNPs are somewhat similar between the two studies, the critical difference lies in how each interprets local adaptation’s extent. Such discrepancy shows the complexity of evolutionary processes in highly dynamic heterogeneous marine environments and the relevance of using convenient genetic markers [45,46]. Given that this debate impacts the management of the species, particularly in defining reproductive, conservation, and management units, the further investigation of their local adaptations is needed.

*Mytilus chilensis* underpins a world-class aquaculture industry, primarily concentrated in the inner sea of Chiloé Island (41° S to 43° S). This industry relies heavily on the availability of seeds artificially collected from natural seedbeds, which are then translocated to ecologically diverse bays until harvest [47]. For instance, Cochamó and Yaldad are two ecologically distinct natural seedbeds located about 250 km apart in the northern and southern zones in the inner sea of Chiloé Island, respectively [41,44]. These locations exhibit a north–south gradient in seawater temperature, currents, salinity, and chlorophyll-a concentration [48,49]. Due to continuous artificial seed extraction and reduced natural recruitment (seed collectors compete with the recruitment of natural beds), some seedbeds have shrunk and show increased inbreeding levels [11]. Translocation and natural larval drift in the water column facilitate hybridization between individuals from different environments, posing a risk of losing locally adapted alleles and eroding genetic diversity [50]. On the other hand, translocated mussels are exposed to a wide range of pathogenic microorganisms [51,52], pollution [53,54], and environmental fluctuations [55,56,57]. These factors negatively affect the growth, health, shell biomineralization, reproductive performance, and larval recruitment [58,59,60,61], ultimately impacting their fitness.

The underlying hypothesis of this study is that seedbeds located in the north (Cochamó in the Reloncaví Fjord) and south part of Chiloé Island (Yaldad) retain local adaptations due to the complex environmental differences, with growth and adaptive gene expression variation distinguishing local from foreign individuals. Additionally, the local selective pressures are expected to counteract genetic homogenization, evidenced by the genomic distribution of the outlier SNPs linked to candidate adaptive genes. To test the hypothesis, a 91-day reciprocal transplant experiment was conducted with individuals from Cochamó (41° S) and Yaldad (43° S), evaluating differential growth and gene expression in control (self-transplanted) and experimental (cross-transplanted) individuals under the δ_LF_ and δ_HA_ criteria. This investigation is expected to enhance our understanding of the genomic adaptive strategies evolved by *M. chilensis* in response to environmental and anthropic perturbations, information required for the efficient conservation and management of local seed-source populations supporting the mussel industry.

## 2. Results

### 2.1. Environmental Characterization

The reciprocal transplant experiment, conducted from 26 April to 28 July 2018, spanned 91 days. Oceanographic conditions registered in Cochamó at the start of the experiment included a seawater surface temperature of 12.2 °C, salinity of 19.0 ppt, and pH of 6.8. In comparison, Yaldad exhibited a seawater temperature of 11.6 °C, 31.4 ppt salinity, and a pH of 7.05. By the end of the experiment, Cochamó’s temperature dropped to 10.4 °C, salinity increased to 20.5 ppt, and pH rose to 7.02. Yaldad’s seawater temperature decreased to 9.6 °C, salinity rose to 32 ppt, and pH to 7.62. Oceanographic data (0 to −10 m) from the CHONOS database (June 2017 to May 2018), coinciding with the period in which the experiment was performed (Figure 1), suggest that Cochamó and Yaldad differ ecologically, with Cochamó exhibiting higher mean seawater temperatures, stronger marine currents, and longer mean water retention times, but lower salinity, than Yaldad. These natural differences have been maintained over a longer period than the duration of the experiment.

### 2.2. Mean Size Comparisons

Comparisons of mean sizes, estimated as weight corrected for length (W*), between the groups (Figure 2)—including local individuals from Cochamó (LCo) and Yaldad (LYa), self-transplanted (ACo, AYa) and cross-transplanted (TCo, TYa)—revealed significant differences (*p*_value_ < 0.005) at both the start and end of the experiment. At the beginning (sampling 1, s1), LCo_s1 individuals were smaller (3.70 ± 0.60 g) than LYa_s1 (5.56 ± 0.63 g). By the experiment’s end, LCo had grown faster (5.03 ± 0.37 g) than LYa (5.93 ± 0.53 g), with an estimated growth rate of 0.015 g/day for LCo and 0.004 g/day for LYa. TCo grew, on average, less (4.54 ± 0.48 g) than LCo and ACo groups (4.83 ± 0.46 g), while TYa (5.91 ± 0.70 g) showed no significant mean size differences from LYa and AYa groups (5.94 ± 0.51 g). The overall mean size across all groups was 5.11 g.

### 2.3. Mappings of Clean Reads

This study compared the number of clean reads obtained after trimming for individuals from Cochamó and Yaldad under self- and cross-transplantation, using the chromosome-level *M. chilensis* genome sequence as a reference (Table 1). Self-transplanted Cochamó individuals (ACo) yielded 54.3 million reads (standard deviation [SD]: 4.1 million), with 69.79% mapped into intergenic regions, 8% mapped into introns, 1.49% into exons, 0.05% into exon–exon junctions, and 20.67% unmapped. Cross-transplanted individuals (TCo) yielded 33.1 million reads (SD = 1.1 million), with 74.50% mapped into intergenic regions, 8.58% into introns, 1.51% into exons, 0.05% into exon–exon junctions, and 15.37% unmapped.

Self-transplanted Yaldad individuals (AYa) yielded 55.6 million clean reads (SD = 677,061), with 69.94% mapped into intergenic regions, 8.04% into introns, 1.50% into exons, 0.04% into exon–exon junctions, and 20.47% unmapped. Cross-transplanted (TYa) individuals yielded 32.85 million reads (SD = 2.9 million), with 74.68% mapped into intergenic regions, 8.52% into introns, 1.47% into exons, and 0.06% into exon–exon junctions, leaving 15.27% unmapped. Across both groups (self- and cross-transplanted) from Cochamó and Yaldad, the highest percentages of reads were mapped into intergenic regions, with a smaller proportion in intronic and exonic regions. Unmapped reads showed similar proportions between conditions but slightly higher in the self-transplanted group.

For self-transplanted individuals from Cochamó (ACo), 54.3 million reads were obtained (standard deviation SD = 4.1 million). Of these, 69.79% mapped to intergenic regions, 8% to intronic regions, 1.49% to exonic, and 0.05% to exon–exon junctions. Unmapped reads made up 20.67% of the total. In the cross-transplanted group, 33.1 million reads were obtained (SD = 1.1 million), with 74.5% mapping to intergenic regions, 8.58% to intronic regions, 1.51% to exonic regions, and 0.05% to exon–exon junctions. Unmapped reads constituted 15.37%. For self-transplanted individuals from Yaldad, 55.6 million clean reads were mapped (SD = 677,061), with 69.94% mapping to intergenic regions, 8.04% to intronic regions, 1.50% to exonic regions, and 0.04% to exon–exon junctions. Unmapped reads made up 20.47%. In the cross-transplanted group, 32.85 million reads were mapped (SD = 2.9 million), with 74.68% mapping to intergenic regions, 8.52% to intronic regions, 1.47% to exonic regions, and 0.06% to exon–exon junctions. Unmapped reads accounted for 15.27%. Overall, across both self-transplanted and cross-transplanted groups from Cochamó and Yaldad, most reads were mapped to intergenic regions, while smaller proportions mapped to intronic and exonic regions. Unmapped reads were similar between groups but higher in self-transplanted individuals.

### 2.4. Differential Expression Analysis

The heat map in Figure 3A highlights regions of high (yellow/red) and low (black) gene expression, clustering samples, and their biological replicates (*p*_value_ ≤ 0.05). Differentially expressed genes were identified, with some highly expressed in Cochamó individuals (ACo and TCo) but not in Yaldad (AYa and TYa), indicating location-specific biological responses. Genes with higher expression in TCo than ACo suggest enhanced expression due to cross-transplantation. ACo and TCo showed distinct patterns from Yaldad but formed a closer cluster, suggesting shared gene regulation or similar environmental responses. Conversely, AYa individuals formed a distinct cluster with unique gene expression profiles. TYa individuals also clustered separately, with some genes showing even higher expression than in AYa, reflecting different responses to varying environmental conditions. The heat map effectively distinguished these samples, and a principal component analysis (PCA) further illustrated gene expression variability, separating samples by replicates and distinguishing groups. In the two-dimensional scatterplot (Figure 3B), principal component 1 and principal component 2 explained 15.3% and 9.4% of the variability, forming four distinct groups consistent with the heat map. The volcano plot (Figure 3C) reinforced these findings, showing up-regulated (red) and down-regulated (blue) genes. While most genes clustered near the origin, several exhibited extreme log_2_(fold change) and high statistical significance (−log_10_(*p*_values_)), indicating their involvement with biological processes triggered by transplantation.

### 2.5. Number of Differentially Expressed Genes (DEGs) by Location

The comparative analysis of the differentially expressed genes (DEGs) in Cochamó and Yaldad individuals under the local vs. foreign (δ_LF_) and home vs. away (δ_HA_) criteria revealed distinctive adaptation patterns (Figure 4A). Under the δ_LF_ criterion, 867 DEGs were identified in Cochamó, while Yaldad exhibited a higher number (942), suggesting more pronounced differentiation in gene expression in Yaldad individuals. Under the δ_HA_ criterion, Cochamó and Yaldad exhibited 927 and 951 DEGs, respectively. Thus, Yaldad individuals consistently showed more DEGs under both criteria, revealing a likely higher sensitivity to habitat change.

The ratio of the sum of fold changes (∑FC) to the number of DEGs (∑N°DEGs) under δ_LF_ and δ_HA_ effectively quantifies gene expression changes per DEG (Figure 4B). Thus, in Cochamó individuals, the higher ratio under δ_LF_ (6.75) than δ_HA_ (6.39) suggests more pronounced gene expression changes when comparing local to foreign individuals than home-to-away environments. Similarly, Yaldad’s higher ratio under δ_LF_ (7.19) than δ_HA_ (6.37) indicates stronger gene expression changes when comparing local to foreign individuals, suggesting a greater specialization of Yaldad’s genome functioning to its local environment. Overall, Yaldad individuals consistently showed higher ratios under both criteria than Cochamó ones, aligning with the previous findings of greater DEG differentiation in Yaldad. The elevated δ_LF_ ratio further highlights genomic adaptation, suggesting that Yaldad organisms are finely tuned to their local conditions.

### 2.6. Venn Diagrams and DEG Selection

Venn diagrams illustrate the number of DEGs in Cochamó and Yaldad individuals, highlighting those with significant fold changes (Figure 5). Using stringent filters (FC_value_ ≥ |4|, FDR *p*_value_ ≤ 0.05), analyses under δ_LF_ and δ_HA_ criteria emphasize exclusive DEGs in each location. The accompanying graphs detail the number and cumulative fold change (∑FC) values for up-regulated DEGs in self- and cross-transplanted individuals. Under the δ_LF_ criterion for Cochamó (Figure 5A), comparing TYa and ACo individuals, 236 DEGs were identified, with 123 exclusives to this comparison. Of these, 37 were up-regulated in ACo (∑FC: 665) and 86 in TYa (∑FC: 1067). In Yaldad (TCo vs. AYa) (Figure 5B), from 298 DEGs identified, 180 were exclusive to this group. Of these, 65 were up-regulated in AYa (∑FC: 557) and 115 in TCo (∑FC: 1142).

Under the δ_HA_ criterion for Cochamó (TCo vs. ACo) (Figure 5C), 258 DEGs were identified, and 153 were exclusive, of which 50 were up-regulated in ACo (∑FC: 520) and 103 in TCo (∑FC: 1232). In Yaldad (TYa vs. AYa) (Figure 5D), 233 DEGs were identified and 130 exclusives, of which 55 were up-regulated in AYa (∑FC: 947) and 75 in TYa (∑FC: 727).

Collectively, these Venn diagrams and their associated graphs provide a comprehensive overview of the significant DEGs differentiating Cochamó and Yaldad individuals, highlighting distinctions under the δ_HA_ and δ_LF_ operational criteria for local adaptations. It should be noted that the estimated TPM values by the RNA-Seq analysis (as a proxy for gene expression patterns) for several genes were validated through relative expression value analyses comparing with their respective estimates by qRT-PCR (Data S1).

### 2.7. Annotation of DEGs

The DNA sequences of several selected DEGs showed homology with known sequences from databases like SwissProt, Pfam, BLAST NR, and eggNOG, enabling nominal assignments to DEGs for each comparison under δ_LF_ and δ_HA_ criteria (Appendix A). These annotations offered valuable insights into the genomic responses of Cochamó and Yaldad individuals to transplants, highlighting the specificity and complexity of their responses.

#### 2.7.1. Local vs. Foreign (δ_LF_) Criterion Comparison

Table 2 details the annotations for the top ten DEGs for Cochamó (TYa vs. ACo comparison) and Yaldad (TCo vs. AYa comparison) under the δ_LF_ criterion. In Cochamó, the first two DEGs in the ACo listed in Table 2A stand out due to their significant increases in expression. The first DEG, MCH017805.1, exhibited an FC_value_ of 216 and likely encodes a T-cell-specific GTP nucleotide protein (SwissProt ID A0A1S3IRL5_LINUN), an interferon-related protein linked to immune response. The heightened expression of this DEG suggests an intensified immune system functioning in ACo. Likewise, MCH006397.1 (FC_value_ of 91) likely encodes a zinc knuckle domain protein (Pfam ID PF00098.22), essential for DNA interaction and gene expression regulation, indicating increased regulatory activity. In the TYa group, the top two DEGs in Table 2A exhibit significant up-regulation. MCH017771.1 (FC_value_ of 74) likely encodes a solute carrier protein (SwissProt ID A0A210QJ59_MIZYE), suggesting enhanced nutrients or molecule transport. MCH033811.1 (FC_value_ of 55) encodes a hypothetical protein (BLAST NR ID OPL20729.1) yet to be fully characterized, likely playing a critical role in TYa individuals’ response to the new environmental conditions.

For the δ_LF_ criterion in Yaldad (TCo vs. AYa comparison), the top two DEGs in AYa (Table 2B) showed significant up-regulation. MCH024186.1 (FC_value_ of 45) encodes a protein involved with cellular signaling pathways (eggNOG ID 126957.SMAR011084), suggesting a heightener of cellular communication and signaling activity. MCH020826.1 (FC_value_ of 40) encodes for a Caprin family protein (eggNOG ID 144197.XP_008303365.1) involved with mRNA transport at the cellular level, cell growth, and thermal stress response. In TCo, the top two DEGs in Table 2B exhibit significant up-regulation. MCH003374.1 (FC_value_ of 59) likely encodes a B-box zinc domain protein (Pfam ID PF00643.23) involved in gene expression regulation and response to extracellular signals, indicating intense genetic expression regulation in TCo. MCH017866.1 (FC_value_ of 55) encodes a protein not fully characterized (BLAST NR ID XP_021372906.1), likely playing a role in TCo response to the new environment.

#### 2.7.2. Home vs. Away (δ_HA_) Criterion Comparison

Table 3 lists the top ten annotated DEGs for Cochamó (TCo vs. ACo comparison) and Yaldad (TYa vs. AYa comparison) under the δ_HA_ criterion. In Cochamó, the top two DEGs in ACo (Table 3A) are notable for up-regulation. MCH016307.1 (FC_value_ of 39) likely encodes the Coagulation factor C-terminal domain (eggNOG ID 10224.XP_006822956.1), essential for tissue repair and homeostasis. MCH014593.1 (FC_value_ of 23) likely encodes Complement C1q-like (BLAST NR ID XP_011447428.1), linked to the complement system activation and innate immune response. In the TCo group, the top two DEGs in Table 3A are also notable for up-regulation. MCH017771.1 (FC_value_ of 109) encodes a solute carrier family 46 protein (SwissProt ID A0A210QJ59_MIZYE), suggesting enhanced nutrients or molecule transport. Likewise, MCH033811.1 (FC_value_ of 97) likely encodes a hypothetical protein (BLAST NR ID OPL20729.1), suggesting that it may play a key role in TCo individuals’ response to new environmental conditions.

In Yaldad, the top two DEGs in AYa in Table 3B showed notable expression increases. MCH023206.1 (FC_value_ of 199) likely encodes interferon-induced transmembrane protein (eggNOG ID PF04505.11), essential for the immune response to infections. MCH018071.1 (FC_value_ of 158) likely encodes zinc knuckle (BLAST NR ID PF00098.22), linked to gene expression regulation. In the TYa group, the top two DEGs listed in Table 3B are notable for their significant up-regulation. MCH006622.1 (FC_value_ of 49) likely encodes Peroxisomal NADH pyrophosphatase (SwissProt ID K1QES1_CRAGI), essential for metabolism and cellular homeostasis. Likewise, MCH025739.1 (FC_value_ of 25) encodes Aquaporin-2 (BLAST NR ID EKC32884.1), which regulates water transport across membranes and maintains cellular osmotic balance.

### 2.8. Functional Categorization of DEGs

Annotations via KOBAS and REVIGO matched KEGG and GO ID terms for DEG sequences identified under δ_LF_ and δ_HA_ criteria. KEGG offered insights into metabolic and signaling pathways. However, it was less informative than GO, which provided a broader spectrum of biological processes, cellular components, and molecular functions related to genomic response to transplantation in *M. chilensis*.

#### 2.8.1. KEGG Categorization

Under the δ_LF_ criterion for Cochamó, none of the 37 DEGs in ACo (TYa vs. ACo) had a functional assignment in the KEGG database (Table 4). In contrast, 7 of 86 DEGs in TYa were assigned to eight KEGG terms, highlighting amino acid and carbohydrate metabolism. For δ_LF_ in Yaldad (TCo vs. AYa), 3 of 65 DEGs in AYa were linked to four KEGG terms, including extracellular matrix receptor interaction and endocytosis. Meanwhile, 4 of 115 DEGs in TCo were assigned to 11 KEGG terms involving lipid and protein metabolism.

Under the δ_HA_ criterion for Cochamó, none of the 50 DEGs in ACo (TCo vs. ACo) found functional assignment; meanwhile, 7 of 103 DEGs in TCo were assigned to 11 KEGG terms involving amino acid, carbohydrate, and lipid metabolism (Table 5). For δ_HA_ in Yaldad, two DEGs in AYa in the TYa vs. AYa comparison were assigned to seven KEGG ID terms linked with amino acid and carbohydrate metabolism. Meanwhile, only two DEGs in TYa were assigned to seven KEGG ID terms related to lipid metabolism.

#### 2.8.2. Gene Ontology (GO) Categorization

Under the δ_LF_ criterion for Cochamó (TYa vs. ACo), 182 GO terms were identified, with 95 terms matching DEGs in ACo and 87 in TYa samples. In ACo, 48% were related to biological processes (BPs), 26% to cellular components (CCs), and 25% to molecular functions (MFs) (Appendix A). Key terms included signal transduction (BP), cytoplasm (CC), and metal ion binding (MF). In TYa, 47% were BP, 24% CC, and 29% MF, with the most frequent terms including cellular nitrogen compound metabolic process (BP), membrane (CC), and nucleic acid binding (MF). Under the δ_LF_ criterion for Yaldad (TCo vs. AYa), 416 GO ID terms were identified, with 182 in AYa and 234 in TCo. In AYa, 71% were related to BP, 12% to CC, and 17% to MF, highlighting the regulation of the DNA-templated transcription (BP), membrane (CC), and metal ion binding (MF). In TCo, 62% were BP, 23% CC, and 15% MF, highlighting the gene expression (BP), membrane (CC), and DNA binding (MF).

Under the δ_HA_ criterion for Cochamó (TCo vs. ACo), 436 GO ID terms were identified, with 91 in ACo samples and 345 in TCo. In ACo, 51% were related to BP, 33% to CC, and 16% to MF (Appendix A). Key terms included signal transduction (BP), plasma membrane (CC), and protein binding (MF). In TCo, 64% were related to BP, 16% to CC, and 20% to MF, highlighting the cellular nitrogen compound metabolic process (BP), membrane (CC), and catalytic activity (MF). Under the δ_HA_ criterion for Yaldad (TYa vs. AYa), 293 GO ID terms were identified, with 170 in AYa and 123 in TYa. In AYa, 62% were BP, 15% CC, and 24% MF. Key terms included regulation of DNA-template transcription (BP), membrane (CC), and metal ion binding (MF). In TYa, 54% were BP, 26% CC, and 20% MF, highlighting the signal transduction (BP), membrane (CC), and ion binding (MF).

The graphical comparison of the 15 most frequently enriched GO ID terms, representing biological processes (BPs) likely involved in the genomic response to transplantation, allowed a deeper exploration of the functional meaning of gene expression differences. Results for δ_LF_ and δ_HA_ are shown in Figure 6 and Figure 7, respectively.

Under the δ_LF_ criterion for Cochamó (Figure 6A), ACo processes included signal transduction, gene expression regulation, and protein ubiquitination, while TYa involved nitrogen metabolism, xenobiotics response, and circadian rhythm. For Yaldad (Figure 6B), AYa processes included transcription and gene expression regulation, ion transport, and cell cycle regulation, whereas TCo involved gene expression regulation and inflammatory and immune responses. Under the δ_HA_ criterion for Cochamó (Figure 7A), ACo included signal transduction, protein transport, and localization, while TCo involved nitrogen metabolism, DNA repair, and oxidative stress response. For Yaldad (Figure 7B), AYa processes involved signal transduction, chromatin remodeling, and the electron transport chain, while TYa included protein transport and modification, transmembrane proton transport, and response to a bacterium.

### 2.9. Genome Mapping of Outlier SNPs and Their Neighboring Genes

#### 2.9.1. Mapping of the Outlier SNPs

Mapping the sequences of 58 outlier SNPs against the chromosome-level genome sequence of *M. chilensis* revealed their distribution in nearly all chromosomes. Figure 8 shows an idiogram illustrating their genomic position (horizontal blue lines) across chromosomes 1 to 14. For example, chromosome 1 (Chr 1) contains seven distinct outlier SNPs visually displayed on this chromosome. Similarly, the other chromosomes (2–14) show varying numbers of outlier SNPs (3, 4, 6, 0, 8, 2, 5, 5, 5, 3, 1, 5, and 4, respectively), providing an organized view of their distribution within the *M. chilensis* genome.

Notably, chromosome 5 (Chr 5) shows no marks, indicating this linkage group’s lack of detected outlier SNPs. In contrast, chromosome 6 (Chr 6) exhibits a higher density of outlier SNPs (8), suggesting greater genetic variability than chromosome 2 (Chr 2), which showed only two outlier SNPs. This comparison highlights the number variation and distribution of outlier SNPs across chromosomes, offering a comprehensive landscape of the genomic differences driven by these adaptive punctual mutations throughout the *M. chilensis* genome.

#### 2.9.2. Annotations of Outlier SNP-Neighboring Genes

By mapping the outlier SNPs within the *M. chilensis* genome sequence, 47 neighboring genes within 20 Kb up- and downstream were identified. Table 6 provides detailed annotations, including chromosomal location, the database used, and a brief description of each gene. For example, outlier SNP 2352_34 on Chr 1 is physically linked to the gene annotated for an uncharacterized protein C19orf41-like. Similarly, outlier SNP 6660_22 on the same Chr 1 was linked to genes for phosphatase regulatory subunit and zinc finger SWIM proteins. On Chr 3, outlier SNP 4790_45 was linked to three genes lacking homology or functional description. On Chr 8, outlier SNP 6819_55 is linked to three genes, including those related to Pogo and the Maelstrom spermatogenic silencer transposons.

Many neighboring genes of the outlier SNPs have well-defined descriptions, such as the centromeric complex protein linked to SNP 836_12 on Chr 4 and the dnaJ protein linked to outlier SNP 2134_59 on Chr 13. However, some genes remain uncharacterized. Interestingly, none of the found outlier SNP-neighbor genes were part of the DEGs detected from differential expression analyses of the reciprocal transplant experiment.

#### 2.9.3. GO Categorization of Outlier SNP-Neighboring Genes

The Gene Ontology (GO) analysis identified 148 terms for the 47 neighboring genes of outlier SNPs: 78 in biological processes (BPs), 30 in cellular components (CCs), and 40 in molecular functions (MFs). Figure 9 shows the top 50 enriched GO terms. In BP, notable processes include protein localization, ubiquitination, translation, DNA repair, cytoskeleton organization, and innate immune response. Other notable processes include signal transduction, DNA transcription, and cell cycle regulation. For CC, key locations of gene products are the extracellular region, chromatin, nucleus and nucleoplasm, lysosome, endoplasmic reticulum, and Golgi membrane. Other important components include plasmatic membranes and protein-containing complexes. In MF, various functions involved ubiquitin-protein transferase, protein kinase, GTPase activities, and various binding activities for zinc, calcium, nucleic acids, and enzymes.

This GO analysis highlights the extensive range of functions and cellular localizations putatively associated with the mapped outlier SNP-neighbor genes, offering valuable insights into their potential roles in fitness-related biological functions. These annotations emphasize the significance of these outlier SNPs in genetic research and provide a deeper understanding of the intricate genomic landscape of *M. chilensis*.

## 3. Discussion

The reciprocal transplant experiment performed using the local vs. foreign (δ_LF_) and home vs. away (δ_HA_) criteria to test local adaptations [12] provided physiologic and genomic evidence that *Mytilus chilensis* individuals from Cochamó and Yaldad, two natural seedbeds located in ecologically distinct zones in the inner sea of Chiloé Island, retain local adaptations. This finding proposes a new scenario to that of the genetic homogenization attributed to high genetic connectivity due to the high dispersal of the planktonic larvae and seed translocations [42,43,62]. Individuals from Cochamó and Yaldad fit the model of populations with gene flow, where local adaptation emerges from the balance between dispersal and the selective pressures exerted locally [1,2,6,12]. The results also highlight the relevance of properly identifying locally adapted seedbeds for conserving and managing *M. chilensis*, particularly in defining reproductive and management units. The following lines discuss major evidence of how native individuals from Cochamó and Yaldad express different adaptive strategies in their local environment and the impact of translocations on the growth rate, differential genic expression, and genome functioning.

### 3.1. Environmental Barriers as Drivers of Local Adaptation in Mytilus chilensis

As said before, the influence of environmental barriers over gene flow in shaping local adaptation in *M. chilensis* is significant and cannot be overlooked [30,34,35,36]. The natural north–south oceanographic barriers in the inland sea of Chiloé Island, manifested in seasonal differences in temperature, salinity, marine currents, water age (Figure 1), and chlorophyll-a abundance [49] over the years, maintain distinct selective environments that impact larval availability and impose contrasting pressures on mussel survival and reproductive performance. These environmental characteristics provide an optimal scenario for investigating local adaptation, where selective pressures act as a filter, favoring traits that confer survival advantages in specific habitats. Previous field and laboratory studies have evaluated the responses of *M. chilensis* to various environmental factors, including temperature [63,64], salinity [33], acidification [58,59], and the presence of different toxins due to toxic algal blooms [65,66,67,68]. Predators also affect mussel survival [32,69,70], further highlighting the complexity and significance of these selective forces.

### 3.2. Growth Rate as Indicator of Local Adaptation in Mytilus chilensis

While demonstrating local adaptation requires direct evidence of fitness differences, the short duration of this transplant experiment limited such assessments. As Kawecki and Ebert (2004) [1] and Savolainen et al. (2013) [2] noted, measuring fitness can be challenging, particularly in short-term or intragenerational studies like this. Nevertheless, *M. chilensis* has demonstrated sensitivity to environmental changes. The transplantation of individuals from the local environment at a 4 m depth to shallower areas (1 m depth), for instance, markedly affects their growth, calcification rates, and metabolic stress levels [15], which are critical fitness indicators. Also, examining fitness-related traits such as morphometric differences, valve shape, or growth rates provides valuable insights into how habitat changes impact these organisms [27]. In particular, the growth rate, as reflected in weight gain, is a useful fitness indicator and offers insights into how well individuals adapt to their specific environments. Faster growth is associated with a higher likelihood of reaching maturity, reproducing successfully, and leaving offspring, suggesting that individuals who grow more rapidly are better adapted, increasing their chances of survival and passing on their genes to future generations [71,72].

In this study, although individuals from Cochamó were initially smaller during the 91-day transplant experiment, they exhibited a significantly higher growth rate (0.015 g/day) than those from Yaldad (0.004 g/day). Furthermore, Cochamó individuals appeared to be more affected by habitat changes, as reflected by the average sizes of self-transplanted and cross-transplanted individuals, which did not reach the overall average across all samples (Figure 2). This observation suggests an adaptive disadvantage when exposed to non-native environments, aligning with the concept that individuals typically exhibit higher fitness in their native habitats [1,2,6,12]. The observed growth rate differences between Cochamó and Yaldad populations likely reflect their evolutionary fine-tuning to specific environmental conditions, with each population performing better in its local environment. This evidence highlights the influence of regional ecological pressures on the fitness of these populations, reinforcing the idea of local adaptation. However, it is relevant to note that morphometric differences between populations of bivalve mollusks, including mussels [73,74], have evolved to historical environmental conditions over multiple generations. Given their sensitivity to contemporary ecological changes—such as those driven by climate change or disturbances in the aquaculture ecosystem of Chiloé Island—transcriptomic data were instrumental in detecting adaptive differences, representing complementary evidence from the genomic level of biological organization.

### 3.3. Differential Transcription Across the Mytilus chilensis Genome

Transcriptomic analyses revealed significant differential gene expression patterns between the Cochamó and Yaldad populations of *M. chilensis* (Figure 3), which reflect distinct molecular mechanisms driving the genomic responses to environmental pressures. This study further unveils that different regions across the *M. chilensis* genome are differentially transcribed. The RNA-Seq data showed significant variations in the proportion of reads mapped to intergenic, intronic, and exonic regions (Table 1). For instance, most reads were mapped into intergenic regions from self- and cross-transplanted individuals from Cochamó and Yaldad, often rich in regulatory elements such as promoters and enhancers that modulate gene expression [75]. An example includes epigenetic factors such as lncRNAs, which were differentially expressed in the genome of local individuals [36], which likely also influence the gene expression differences observed in transplanted individuals. On the other hand, it is known that the regulation of gene expression is the fundamental link between the genotype, phenotype, and environment, whose variation is recognized as a major source of adaptive evolution [76]. In this context, a smaller percentage of reads was mapped to intronic and exonic regions, suggesting that transcribed protein-coding sequences are less abundant. This finding is consistent with the high proportion of repetitive sequences present in the genome [30]. For instance, heterochromatin contains repetitive sequences, plays a relevant regulatory role in the genome, and is linked with epigenetic factors [77,78].

Moreover, the higher proportion of unmapped reads in self-transplanted individuals, along with a lower proportion of reads in intronic regions, may be attributed not only to the presence of local environment-specific transcripts not represented in the reference genome sequence but also to more efficient splicing and pre-RNA maturation processes. These processes are crucial for proper gene expression and cellular function and critical to adaptive divergence [76,79,80,81]. Thus, these findings underscore the complexity of the genomic adaptation of *M. chilensis*, where specific regions of the genome are fine-tuned to meet the demands of local environmental pressures.

### 3.4. Genomic Differentiation and Local Adaptation in Mytilus chilensis

Local adaptations are reflected in genetic differentiation between populations, driven by selective pressures from the local environment [2,82]. In this study, individuals from Cochamó and Yaldad displayed distinct genome expression strategies. Metrics such as the total number of differentially expressed genes (ΣN°DEGs) and the ratio between the total sum of fold change (ΣFC) and ΣN°DEGs (ΣFC/ΣN°DEGs), evaluated under the δ_LF_ and δ_HA_ criteria, revealed that Cochamó individuals exhibited fewer DEGs and minor gene expression changes in response to transplantation compared to those from Yaldad. In Yaldad, more genes showed significant expression changes, accompanied by a higher ΣFC/ΣN° DEG ratio (Figure 4). Suppose that ΣN°DEGs indicates the magnitude of the genomic adaptive response to an environmental change and ΣFC/ΣN°DEGs approximates the expression intensity per DEG. In that case, a high ratio combined with many DEGs suggests that numerous genes significantly contribute to adaptive change. These patterns likely indicate strong selective pressures and adaptation to the local environment, as seen in Yaldad, where the increased ratios and DEGs under the δ_LF_ criterion suggest that organisms likely invested more energy (e.g., ATP) to maintain their specific adaptation into the foreign environment. Likely, neutral SNPs cannot detect such specific gene changes in response to the environment.

Conversely, the lower ratios and fewer DEGs observed in Cochamó individuals may reflect less intense selection, potentially due to a less demanding environment or gene flow with nearby locations [44]. However, under the δ_HA_ criterion, Cochamó shows a higher number of DEGs, indicating that organisms are expending more energy to adapt to foreign conditions, reflecting a physiologically costly response in terms of plasticity. This observation could explain the reduced growth observed in transplanted individuals. Thus, differential gene expression in response to local or foreign environments can also be linked to the energy costs associated with adaptation. Organisms often face trade-offs between adapting to their native conditions and responding to new or stressful environments, which can result in physiological costs. These findings align with the expectation that natural selection favors adaptive advantages in a particular environment.

### 3.5. Identifying Candidate Adaptive DEGs in Mytilus chilensis for Local Adaptation

The identification of DEGs (Table 2 and Table 3), many of which are regulatory genes linked to functions such as solute transport, cell signaling, and immune response (Figure 6 and Figure 7), suggests that different regulatory biological processes, cellular components, and molecular functions are involved in the adaptive responses of *M. chilensis*. The differential expression of these regulatory DEGs indicates that gene expression advantageous in one environment may not necessarily be beneficial in another. In this study, significant changes were observed in the expression of genes related to immune response, osmoregulation, stress response, thermal regulation, and oxygen concentration management, among other fitness-related traits. For instance, the gene coding for a C1q-like complement protein was differentially expressed in self-transplanted individuals from Cochamó. The genic product of the C1q gene plays a crucial role in activating the complement system in innate immunity and broader regulatory functions, including the modulation of immune tolerance and the inflammatory response [83,84,85].

Similarly, the gene coding for a member of solute carrier family 46 plays a crucial role in transporting essential solutes across cell membranes, a function vital for maintaining osmotic balance in environments with fluctuating salinity levels [82]. Additionally, the up-regulation of Aquaporin, which is involved in water transport across cell membranes, suggests adaptation to varying salinity levels. In Yaldad, higher salinity requires efficient water regulation, which could be energy-optimized for these conditions. While optimized for Yaldad, this adaptation might be disadvantageous in the lower-salinity environment of Cochamó, potentially leading to excessive dehydration or wasted energy. This mechanism of osmoregulation is comparable to the role Aquaporins play in species like the quagga mussel (*Dreissena rostriformis*), where Aquaporins are crucial for their reproductive success in freshwater environments [86], as they help manage water influx, cellular osmoregulation, and survival.

Another example is the Caprin gene, which regulates cell growth and thermal stress response [87,88] and showed differences in expression. Self-transplanted individuals from Yaldad (with a lower seawater temperature than Cochamó) showed the up-regulation of this gene. Oxygen concentration is also critical for the survival of *M. chilensis* [89]. In Cochamó, where oxygen levels may be lower due to older water [90], higher gene expression encoding a zinc knuckle domain-containing protein involved in hypoxia stress response [91] was observed. This gene could regulate the expression of other candidate genes to improve oxygen utilization efficiency under hypoxia, enabling the mussels from Cochamó to survive in an environment with lower oxygen availability. These findings reinforce that local adaptation in *M. chilensis* is a multi-systemic phenomenon involving evolutionary trade-offs necessary for genomic changes for survival in specific environments.

### 3.6. Mapping of Outlier SNP-Neighbor Genes in the Mytilus chilensis Genome

In addition to investigating the location-specific genome functioning, this study mapped previously published 58 outlier SNPs [44], which were interpreted as signs of local adaptation. These outlier SNPs were linked to key neighboring genes annotated for essential functions in regulating solute transport, oxidative stress response, and oxygen management, among other fitness-related functions (Table 6, Figure 9). Likewise, the presence of hundreds of location-specific monomorphic genetic variants (f > 0.99), as also reported in the whole [34] and mitochondrial [35] transcriptomes of local individuals from Cochamó and Yaldad, further highlights the importance of these genetic markers in local adaptation [20]. These findings underscore the uneven distribution of selective signatures across the genome. For instance, chromosome 1 exhibited several mapped outlier SNPs, while chromosome 5 mapped none. This observation supports the idea that selective marks are not uniformly distributed throughout the genome and that natural selection strongly targets specific chromosomal regions [92].

Overall, the comprehensive analysis presented in this study supports the hypothesis that natural selection drives local adaptation in *M. chilensis* and underscores the value of integrating functional genomic studies into conservation and management efforts. The results obtained from growth rate, transcriptomic, and outlier SNP-neighbor adaptive gene analyses provide new insights into the mechanisms underlying local adaptation in this species, offering a solid foundation for future research to test local adaptations. Although these molecular analyses provide a powerful tool for understanding adaptive processes, it is essential to complement these findings with functional studies such as cloning, knockout, CRISPR/Cas9, or overexpression experiments—to confirm the adaptive roles of the identified candidate adaptive DEGs, the outlier SNP-neighboring genes, and monomorphic transcriptomic variants in local adaptation. These approaches would help validate the significance of the identified candidate adaptive genes in responding to the specific environmental conditions of Cochamó and Yaldad, thereby demonstrating their contribution to fitness and survival in these environments. However, these findings not only enhance our understanding of the adaptive biology of this species but also carry important practical implications for marine resource management and biodiversity conservation, particularly in regions highly perturbed, like the inner sea of Chiloé Island. Insights into local adaptations can guide management strategies for optimizing the productivity and sustainability of *M. chilensis* populations. Furthermore, the evidence of local adaptation highlights the critical need for cautious management of mussel transplant practices between farming sites to preserve these adaptations, essential for maintaining productivity and resilience against environmental challenges, especially in the face of climate change.

## 4. Materials and Methods

### 4.1. Study Sites and Sampling

The inner sea of Chiloé Island exhibits notable oceanographic differences between its northern and southern zones, such as Reloncaví Fjord (41.5° Lat S) and Corcovado Gulf (43.5° Lat S), respectively. These disparities are shaped by seasonal fluctuations driven by climatic factors, marine currents, and water conditions. In the northern zone, mean salinity values of 21.2 ± 1.86 ppt and sea surface temperatures of 17 ± 0.72 °C have been recorded in summer, dropping to 11.3 ± 0.45 °C in winter. In contrast, the southern zone experiences salinity levels of 31.6 ± 0.46 ppt and temperatures of 21.7 ± 5.16 °C, decreasing to 11.5 ± 1.06 °C in winter [52].

Native *M. chilensis* individuals were from two distinct seedbeds: Cochamó, at the northernmost tip of the inner sea of Chiloé Island (41°28′23.77″ S, 72°18′38.61″ W), an estuarine bay with a constant influx of freshwater; and Yaldad, at the southernmost tip (43°07′14.63″ S, 73°44′25.72″ W), a coastal bay influenced by open sea currents from Guafo’s mouth. Oceanographic data, including temperature (°C), currents (m/s), salinity (psu), and seawater age (days), were obtained from the CHONOS database (http://chonos.ifop.cl/, accessed on 18 November 2020), managed by the Chilean Institute of Fisheries Enhancement (IFOP). Data were collected from Cochamó and Yaldad at depths ranging from 0 to −10 m, covering June 2017 to May 2018, coinciding with the sampling dates of the reciprocal transplant study. The data were processed, projected, and visualized using Ocean Data View ODV v5.32 software (Reiner Schlitzer, Alfred Wegener Institute for Polar and Marine Research, Bremerhaven, Germany).

### 4.2. Reciprocal Transplant Experiment

The reciprocal transplant experiment started on 26 April 2018, by collecting 600 healthy adult native mussels (e.g., showing siphon activity) in Cochamó and Yaldad. The mussels from each location were approximately 2.7 and 2.9 years old, respectively. Three experimental groups were established at each site: locals (L), self-transplanted (A), and cross-transplanted (T). The local group (L) consisted of 200 individuals, and previous studies have reported comparisons, including analyses of differential gene expressions in whole [34] and mitochondrial [35] transcriptomes, as well as epigenetic factors such as lncRNAs [36].

The self-transplanted (A) group included 200 mussels collected and returned to their original location after being temporarily removed from the water, serving as a control for the transplantation process without the added variable of a new environment. The cross-transplanted (T) group (200 individuals) was relocated, mussels from Cochamó were transplanted to Yaldad, and vice versa. Each group was divided into four replicates of 50 mussels each, housed in plastic mesh cages measuring 35 cm × 25 cm × 15 cm with a mesh opening of 0.5 cm. The cages were submerged at depths of −4 to −8 m, considered a comfort zone for these mussels [15].

The transplant experiment lasted 91 days, culminating on 28 July 2018, when the cages were retrieved and the mussels were collected for measurements. Shell length, width, and thickness were recorded using a caliper, and the mussels were weighed with an analytical balance. Weight measurements were taken after opening the shells and removing excess water. The relationship between weight and shell length was estimated using the ‘powertransform’ function in R (v4.3.3) as a proxy for mean size. Comparisons were made between local individuals at the start (sampling 1, s1) and the end of the experiment among local, self-transplanted, and cross-transplanted individuals. Normalized weight values (g) (W*, *λ* = 0.278) were used to statistically test size differences through ANOVA, followed by post hoc Bonferroni and Tukey tests. At the end of the experiment, all mussels were rinsed with local seawater and stored in separate sterile bags at 10 ± 2 °C before being transported to the University of Los Lagos laboratory for gill tissue sample collection from apparently healthy individuals (e.g., no parasites observed). These samples were stored in cryotubes containing 1 mL of an EZNA^TM^ RNA-Lock Reagent (OMEGA BioTek, Norcross, GA, USA) and were preserved at −80 °C within 4 h of collection.

### 4.3. Taxonomic Affiliation

Thirty individuals from Cochamó and Yaldad were randomly selected, including local, self-transplanted, and cross-transplanted samples. DNA was extracted from gill tissue using the EZNA^TM^ Tissue DNA kit (OMEGA BioTek), following the manufacturer’s instructions. Two independent RFLP assays confirmed taxonomic affiliation as *Mytilus chilensis* [93,94,95]. The first assay amplified 233 pb of the mitochondrial COI gene, digested with Xba I, which cut only in *M. chilensis*. The second assay targeted the nuclear marker Me15/Me16, producing amplicons specific to *M. edulis* (180 bp) and *M. galloprovincialis*/*M. chilensis* (126 bp). Aci I has a restriction site only in *M. edulis*/*M. galloprovincialis*. RFLP results from both genetic markers (COI and Me15/Me16) confirmed that all analyzed individuals were *M. chilensis* (Appendix A).

### 4.4. RNA Extraction and Sequencing

Total RNA was individually isolated from gill tissue using TRIZOL (Invitrogen^TM^, Carlsbad, CA, USA) following the manufacturer’s protocol. RNA integrity was assessed by electrophoresis on 1.2% agarose gels and verified with a TapeStation 2200 system (Agilent Technologies^TM^, Santa Clara, CA, USA). Purity and concentration were measured using spectrophotometry and fluorescence, selecting RNA samples with 260/280 and 260/230 ratios ≥ 2.0 and RIN > 9. Fifteen individual RNA extractions per group (self- and cross-transplanted) were pooled into three biological replicates per location (five individual RNA extractions each). These pools were precipitated overnight in 2 volumes of absolute ethanol and 0.1 volumes of 0.3 M sodium acetate at −80 °C. cDNA libraries were constructed from these pooled RNA samples using the TrueSeq Stranded mRNA LT Sample Prep Kit (Illumina, San Diego, CA, USA), generating 12 high-quality cDNA libraries representing three biological replicates for self- and cross-transplanted individuals at each location. These libraries were sequenced using the Illumina HiSeq 4000 platform (Illumina, San Diego, CA, USA) with a 100-paired-end approach.

### 4.5. RNA-Seq and Differential Expression Analysis

Raw RNA-Seq data were processed using the CLC Genomic Workbench (CLCgw) v24.0.1 (Qiagen Bioinformatics^TM^, Hilden, Germany) to obtain clean reads by trimming adapters, removing low-quality sequences (score threshold of 0.05), and eliminating ambiguous nucleotides and homopolymers from the 3′ and 5′ ends. The same software was used for mapping, normalizing, and quantifying the clean reads with tools from the RNA-Seq analysis suite. Gene expression levels were estimated as transcripts per million (TPMs) values by globally aligning the reads to the chromosome-level *M. chilensis* genome sequence (GenBank BioProject PRJNA861856) [30]. Various filters were applied during read mapping to ensure robustness and minimize biases in genome alignment, where genes and transcripts were annotated. These filters included a mismatch cost of 2, insertion/deletion cost of 3, length/similarity fractions of 0.8, and a 10-hit limit per read. Transcripts with invalid values or zero read counts were excluded. Differential expression was analyzed using a negative binomial generalized linear statistical model (GLM), with the Wald test applied to assess whether differences significantly deviated from zero. Fold change values (FC_value_) were estimated using the GLM to correct for differences in library size and account for biological replicates.

Two filtering approaches were used to explore differential gene expression. Initially, lenient thresholds (FC_value_ ≥|2|, *p*_value_ at ≤0.05) were applied to minimize the risk of Type I errors and assess variability in gene expression differences. Results from the 12 RNA-Seq libraries were visualized through a clustered heat map, organized by replicate (A and T) and location, using Euclidean distances for clustering. Differential gene expression was further assessed with a principal component analysis and volcano plot represented by the relationship −log_10_(*p*_value_) versus log_2_(FC_value_). Thus, for each location, the number of differentially expressed genes (DEGs) passing lenient filters was counted and plotted under both δ_LF_ and δ_HA_ criteria. The ratio of the sum of fold changes (ΣFC) to the total number of DEGs (ΣN°DEGs) was calculated to measure the overall magnitude of gene expression changes per DEG. The number of DEGs reflects the breadth of adaptive response, while the ΣFC/ΣN°DEG ratio indicates the intensity of adjustments per DEG. These metrics provided a comprehensive view of the population’s adaptive genomic strategy.

Subsequently, a more stringent threshold (FC_value_ ≥|4|, FDR adjusted *p*_value_ ≤ 0.05) was applied to reduce false positives from multiple comparisons, focusing on DEGs with significant fold changes. While this filter highlights DEGs with notable differences, it may exclude biologically relevant DEGs with lower fold changes. Identifying these DEGs can be challenging due to similar expression levels with less impactful genes. Alternative approaches like cloning help ensure that relevant DEGs are not overlooked.

Venn diagrams were used to compare samples and identify DEGs that met stringent filtering criteria. FC_value_ were compared based on the δ_LF_ and δ_HA_ criteria, with self-transplantation from Cochamó versus Yaldad (ACo vs. AYa) as the reference. Under δ_LF_, comparisons were made between cross-transplanted Yaldad and self-transplanted Cochamó (TYa vs. ACo) and between cross-transplanted Cochamó and self-transplanted Yaldad (TCo vs. AYa). Additionally, under δ_HA_, comparisons were between cross-transplanted and self-transplanted individuals from both Cochamó (TCo vs. ACo) and Yaldad (TYa vs. AYa). DEGs that passed their filters were identified, annotated, and functionally categorized.

### 4.6. DEG Annotations and Functional Categorization

Different databases, including SwissProt, Pfam, BLAST NR, and eggNOG, were utilized to annotate the selected DEGs in each comparison according to the δ_LF_ and δ_HA_ criteria. DEGs were functionally categorized using the Kyoto Encyclopedia of Genes and Genomes (KEGG) metabolic pathways and Gene Ontology (GO) to explore the potential biological functions. Thus, DEG sequences were functionally categorized using an enrichment analysis with a hypergeometric distribution model on the KOBAS online server [96]. KEGG terms were obtained using *Crassostrea gigas* as a reference, and the GO enrichment analysis was performed with default settings. GO ID terms were refined using the REVIGO online server [97] and Fisher’s exact test to assess the over-representation of GO terms. It aimed to identify the most specific GO ID terms related to biological processes, cellular components, and molecular functions. Semantic graphs highlighted the most enriched GO ID terms across biological processes, providing insights into the putative functional roles of DEGs in both self- and cross-transplanted samples from each location.

### 4.7. Outlier SNP Genome Mapping

The genomic positions of the 58 outlier SNPs, identified by Araneda et al. (2016) [44] as potential signs of local adaptation, were determined by mapping their sequences against the whole-genome sequence [30] of this species using CLCgw software. A global alignment with a length/similarity fraction of 0.8 and a mismatch cost of 2 was applied. The resulting SAM files were uploaded to the GALAXY online server [98], converted into interval, BED, and GFF formats, and re-uploaded to CLCgw for further annotations. The extract annotations tool identified genes flanking up to 20 Kb upstream and downstream of the outlier SNPs. These neighboring genes were then annotated and functionally categorized, providing insights into their potential roles in adaptation.

## 5. Conclusions

1. Two ecologically contrasting *Mytilus chilensis* seedbeds in the north and south of Chiloé Island, respectively, exhibited patterns of local adaptation as demonstrated by this reciprocal transplant study and the used criteria (δ_LF_ and δ_HA_), which suggests that individuals from each location have evolved genomic traits essential for survival and fitness in their native environments. Mussels thrive in their native environments, but experience reduced growth when transplanted, indicating an adaptive disadvantage outside their native habitat.

2. The analyses of differential gene expression and the identification of outlier SNP-neighboring candidate adaptive genes underscore these putative genomic traits’ roles in fitness-related processes such as osmoregulation, immune response, and oxidative stress management. These processes are likely relevant to the adaptive strategies that enable *M. chilensis* to succeed in specific environments, coping with stressors like temperature, salinity, oxygen concentration changes, and pathogen presence.

3. Understanding these local adaptations and other evolutionary drivers, such as stochastic processes (e.g., bottlenecks, founder effect) resulting from larval drift in the water column and seed transplantations, is essential for optimizing aquaculture practices. Aligning management strategies with the specific needs of different natural populations ensures both the sustainability and productivity of *M. chilensis* farming at a regional and global scale.

4. Special attention must be given to monitoring the impact of artificial seed collections on larval grounds, as systematic seed extraction could jeopardize natural recruitment and threaten the persistence of wild populations. By integrating these site-specific genomic insights with sustainable management practice, there is great potential to promote the long-term health and viability of *M. chilensis* aquaculture while, at the same time, conserving—or, when necessary, restoring—these natural seedbeds.

## Figures and Tables

**Figure 1 ijms-26-00931-f001:**
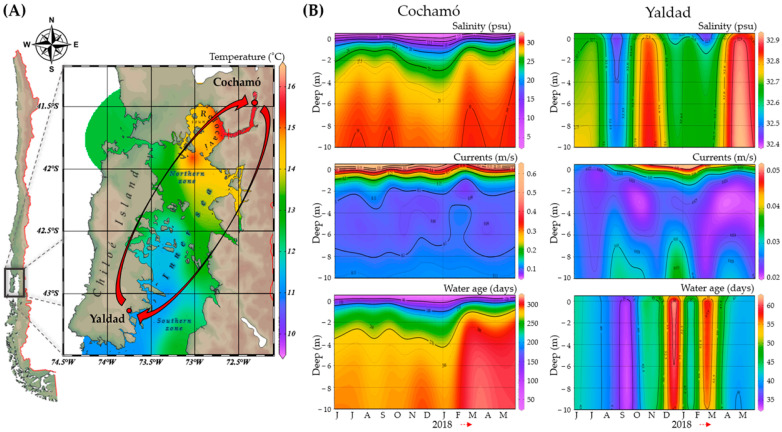
A map illustrating the average temperature (**A**), salinity, currents, and water age (**B**) within the upper 10 m depth of Chiloé Island’s inner sea from June 2017 to March 2018. The map also highlights the locations of two natural *Mytilus chilensis* seedbeds: Cochamó in the north and Yaldad in the south. The red arrows in (**A**) represent the reciprocal transplant experiment.

**Figure 2 ijms-26-00931-f002:**
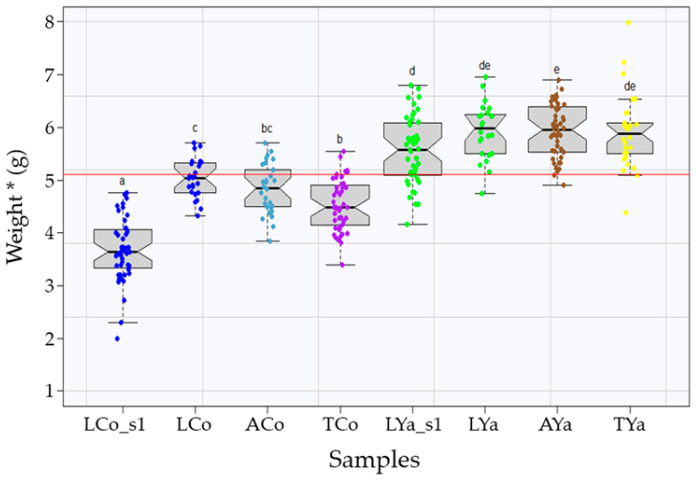
The comparison of mean weights (Weight*) among local (L), self-transplanted (A), and cross-transplanted (T) individuals from Cochamó (Co) and Yaldad (Ya). “_s1” indicates the first sampling. The red line shows the overall mean (5.11 g), and lowercase letters denote statistically significant differences. The estimated growth rates were 0.015 g/day for LCo and 0.004 g/day for LYa. The colors in the graph indicate samples from local (blue, LCo), self-transplanted (light blue, ACo), and transplanted individuals (purple, TCo) from Cochamó, as well as local (green, LYa), self-transplanted (brown, AYa), and transplanted individuals (yellow, TYa) from Yaldad.

**Figure 3 ijms-26-00931-f003:**
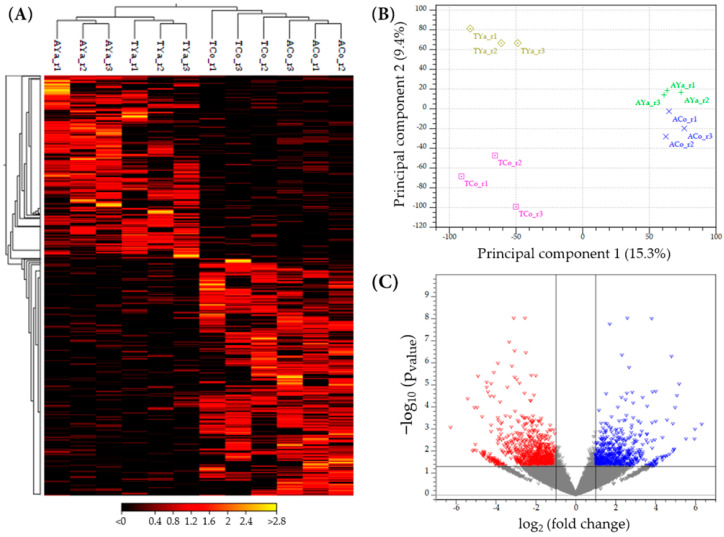
(**A**) A heat map showing differential gene expression by replicate in self- and cross-transplanted individuals from Cochamó (ACo, TCo) and Yaldad (AYa, TYa). The scale bar represents fold change, with colors ranging from black (low expression) to red (high expression). _r1, _r2, and _r3 indicate biological replicates 1, 2, and 3. (**B**) A PCA plot derived from the principal component analysis of replicated samples from Cochamó and Yaldad. ACo and AYa represent self-transplanted individuals from Cochamó and Yaldad, respectively, while TCo and TYa correspond to cross-transplanted individuals between these locations. (**C**) A volcano plot pointing out genes with significantly different expression levels between Cochamó and Yaldad samples. Red triangles indicate up-regulated genes, while blue triangles indicate down-regulated genes. Most genes cluster near the origin (grey triangles), reflecting minimal expression changes, while genes with extreme log_2_(fold change) reveal substantial differences in gene expression.

**Figure 4 ijms-26-00931-f004:**
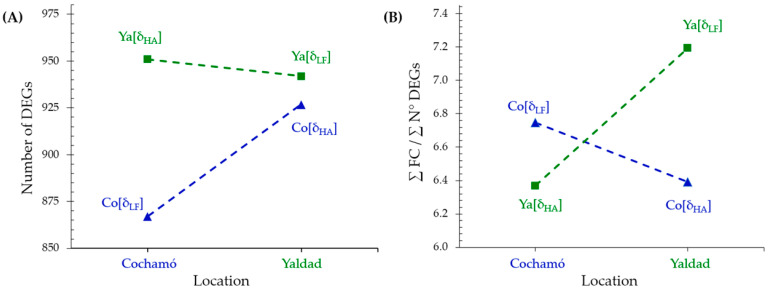
Comparison between (**A**) the number of DEGs and (**B**) the ratio of the sum of fold change to the total number of DEGs (ΣFC/ΣN°DEGs) in Cochamó (Co) and Yaldad (Ya) under the criteria local vs. foreign (δ_LF_) and home vs. away (δ_HA_) for testing local adaptation. Green squares represent Yaldad, and blue triangles represent Cochamó.

**Figure 5 ijms-26-00931-f005:**
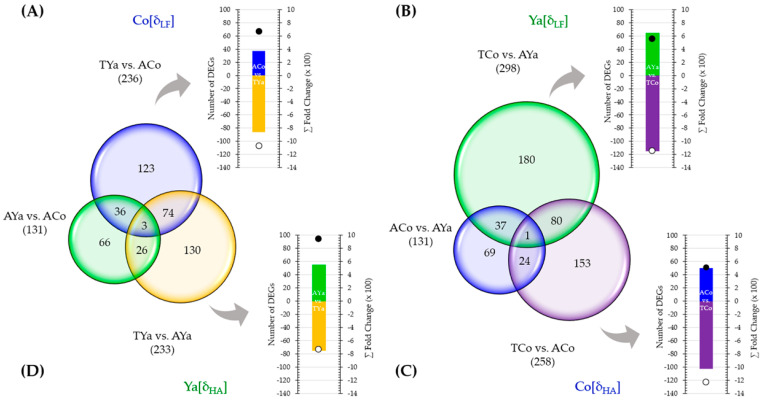
Venn diagrams show the total number of differentially expressed genes (DEGs, in parentheses), including exclusive and shared DEGs, across sample comparisons. These comparisons used the *local* vs. *foreign* (δ_LF_) criterion for Cochamó (**A**) and Yaldad (**B**) and *home* vs. *away* (δ_HA_) criterion for Cochamó (**C**) and Yaldad (**D**). The accompanying bar charts present the number of exclusive DEGs (right axis) and the sum fold change (∑FC, left axis) for each sample, with transplanted samples as negative values. Colors indicate self-transplanted samples from Cochamó (ACo, blue) and Yaldad (AYa, green) and transplanted from Cochamó (TCo, purple) and Yaldad (TYa, yellow).

**Figure 6 ijms-26-00931-f006:**
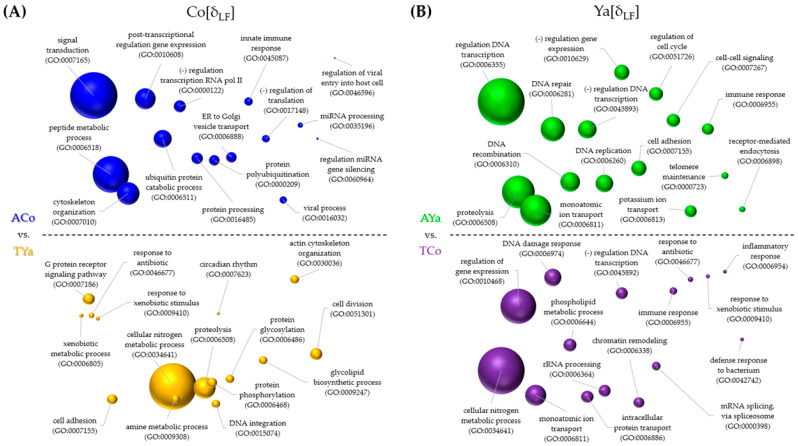
Semantic plots illustrating Gene Ontology (GO) terms enriched under the *local* vs. *foreign* (δ_LF_) criterion in Cochamó (**A**) and Yaldad (**B**). The size of each bubble represents the significance of enrichment, with blue representing self-transplanted from Cochamó (ACo), green representing self-transplanted from Yaldad (AYa), purple for transplanted Cochamó (TCo), and yellow for transplanted Yaldad (TYa).

**Figure 7 ijms-26-00931-f007:**
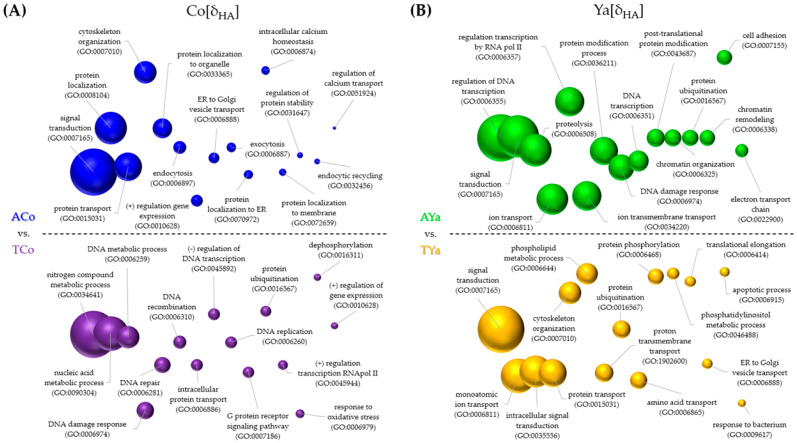
Semantic plots illustrating Gene Ontology (GO) terms enriched under the *home* vs. *away* (δ_HA_) criterion in Cochamó (**A**) and Yaldad (**B**). The size of each bubble represents the significance of enrichment, with blue representing self-transplanted from Cochamó (ACo), green representing self-transplanted from Yaldad (AYa), purple for transplanted Cochamó (TCo), and yellow for transplanted Yaldad (TYa).

**Figure 8 ijms-26-00931-f008:**
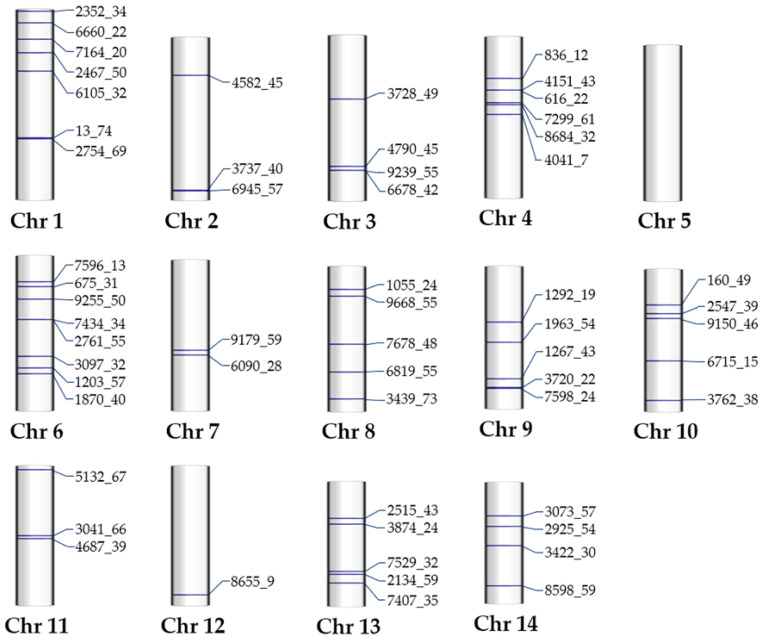
The chromosomal distribution of 58 outlier SNPs across 14 *Mytilus chilensis* genome chromosome sequences. Each chromosome is labeled with the corresponding outlier SNP IDs, illustrating the physical locations of these genetic variants, which may be associated with local adaptation in this species.

**Figure 9 ijms-26-00931-f009:**
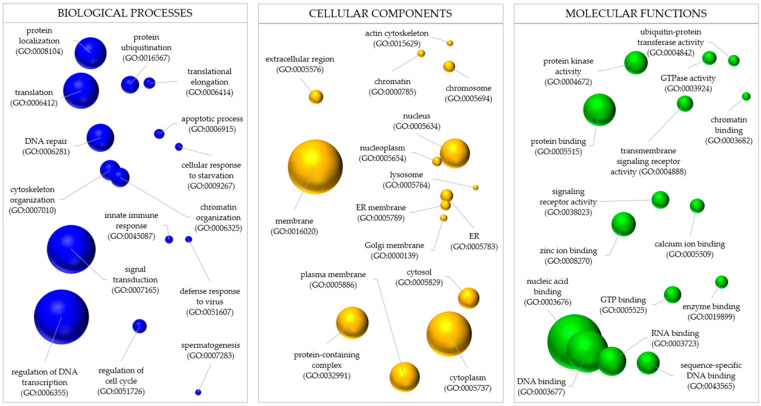
Gene Ontology (GO) categorization for neighbor genes of 58 outlier SNPs mapped in the chromosome-level *Mytilus chilensis*’s genome sequence, organized into three main categories: biological processes (BPs), cellular components (CCs), and molecular functions (MFs). Bubble size represents the frequency of enrichment for each GO term, with colors corresponding to different functional categories: blue for BP, yellow for CC, and green for MF.

**Table 1 ijms-26-00931-t001:** Summary of mean values, standard deviations (SDs), and percentage (%) of mapped reads from Illumina RNA-Seq data for self- and cross-transplanted *Mytilus chilensis* individuals from two natural seedbeds in the inner sea of Chiloé Island: Cochamó (A) and Yaldad (B).

**(A)**		**COCHAMÓ**					
		**Self-Transplanted**			**Cross-Transplanted**		
**Parameter**	**Location**	**Media**	**SD**	**%**	**Media**	**SD**	**%**
Total of reads		54,260,671	4,114,336	100.00	33,079,882	1,061,904	100.00
Reads mapped in unique fragments	Intron	4,341,568	330,607	8.00	2,836,805	102,368	8.58
	Exon	809,780	57,644	1.49	498,607	13,532	1.51
	Exon–exon	24,695	2,749	0.05	16,996	516	0.05
	Intergenic	37,866,782	3,219,598	69.79	24,642,892	724,681	74.50
Reads not mapped		11,217,845	644,857	20.67	5,084,581	333,123	15.37
**(B)**		**YALDAD**					
		**Self-Transplanted**			**Cross-Transplanted**		
**Parameter**	**Location**	**Media**	**SD**	**%**	**Media**	**SD**	**%**
Total of reads		55,553,941	677,061	100.00	32,849,423	2,932,648	100.00
Reads mapped in unique fragments	Intron	4,465,683	119,094	8.04	2,800,303	200,794	8.52
	Exon	831,013	20,365	1.50	483,324	41,619	1.47
	Exon–exon	24,864	1,313	0.04	17,286	1,209	0.05
	Intergenic	38,862,256	809,785	69.95	24,531,770	2,111,543	74.68
Reads not mapped		11,370,125	280,989	20.47	5,016,740	613,594	15.27

**Table 2 ijms-26-00931-t002:** Top ten up-regulated (Up-Reg) differentially expressed genes (DEGs) in each sample comparison under the local vs. foreign (δ_LF_) criterion for Cochamó (A) and Yaldad (B). DEGs were annotated using diverse databases (SwissProt, pfam, BLAST NR, eggNOG) and the species’ whole-genome sequence (GenBank BioProject no. PRJNA861856). Fold change values (FC_value_) of transplanted individuals are presented as negative numbers.

	Comparison	Up-Reg	DEG ID	FC_value_	Database	Database ID	Description
**(A**)	TYa vs. ACo	ACo	MCH017805.1	216.33	SwissProt	A0A1S3IRL5_LINUN	T-cell-specific GTP nucleotide protein
			MCH006397.1	91.16	pfam	PF00098.22	Zinc knuckle
			MCH020394.1	43.56	SwissProt	K1QDR7_CRAGI	Metallo-beta-lactamase domain
			MCH006871.1	39.41	BLAST NR	XP_021346654.1	Uncharacterized protein LOC110446034
			MCH029407.1	30.48	-	-	-
			MCH018328.1	29.46	-	-	-
			MCH027090.1	14.57	eggNOG	7739.XP_002593795.1	Ribonuclease H protein
			MCH020558.1	13.57	-	-	-
			MCH016839.1	12.61	-	-	-
			MCH009573.1	11.80	pfam	PF13833.5	EF-hand domain pair
		TYa	MCH017771.1	−73.63	SwissProt	A0A210QJ59_MIZYE	Solute carrier family 46 member 3
			MCH033811.1	−55.39	BLAST NR	OPL20729.1	Hypothetical protein AM593_09521
			MCH019286.1	−48.22	-	-	-
			MCH000986.1	−48.01	pfam	PF00386.20	C1q domain
			MCH023306.1	−37.26	BLAST NR	XP_011445537.1	Late secretory pathway protein AVL9
			MCH032954.1	−34.72	SwissProt	A0A210PU25_MIZYE	Carboxylic ester hydrolase
			MCH026984.1	−33.34	BLAST NR	XP_022326523.1	Uncharacterized protein LOC111126294
			MCH025896.1	−31.24	-	-	-
			MCH032866.1	−29.60	BLAST NR	OPL20628.1	Collagen chain alpha-1, partial
			MCH024575.1	−25.32	-	-	-
**(B)**	TCo vs. AYa	AYa	MCH024186.1	44.93	eggNOG	126957.SMAR011084	Intracellular signal transduction
			MCH023948.1	39.73	eggNOG	144197.XP_008303365.1	Caprin family member 2
			MCH018636.1	20.41	eggNOG	6412.HelroP185617	Chymotrypsin-like elastase family
			MCH017912.1	19.65	BLAST NR	XP_022336752.1	Glycolipid transfer protein-like
			MCH024803.1	18.35	eggNOG	6500.XP_005107509.1	Zinc ion binding
			MCH028910.1	15.34	-	-	-
			MCH001759.1	13.44	BLAST NR	XP_021343198.1	Proton myo-inositol cotransporter
			MCH023151.1	13.32	BLAST NR	XP_021339691.1	LARGE xylosyl glucuronyltransferase
			MCH018876.1	12.92	-	-	-
			MCH019654.1	11.15	BLAST NR	XP_006817019.1	Uncharacterized protein LOC102808054
		TCo	MCH003374.1	−58.66	pfam	PF00643.23	B-box zinc finger
			MCH017866.1	−54.50	BLAST NR	XP_021372906.1	Uncharacterized protein LOC110462954
			MCH009251.1	−44.54	BLAST NR	XP_002585874.1	Hypothetical BRAFLDRAFT_110974
			MCH004849.1	−39.00	eggNOG	6412.HelroP193608	HEPN domain
			MCH033594.1	−37.98	eggNOG	6500.XP_005107641.1	Peptidase activity
			MCH000114.1	−30.53	eggNOG	136037.KDR22711	Interleukin-like EMT inducer
			MCH003442.1	−26.55	pfam	PF17517.1	IgGFc binding protein
			MCH003165.1	−24.15	pfam	PF01885.15	RNA 2′-phosphotransferase family
			MCH019043.1	−23.81	pfam	PF00098.22	Zinc knuckle
			MCH008976.1	−23.63	SwissProt	U5PYN6_CRAGI	Toll-like receptor 3

**Table 3 ijms-26-00931-t003:** Top ten up-regulated (Up-Reg) differentially expressed genes (DEGs) in each sample comparison under the home vs. away (δ_HA_) criterion for Cochamó (A) and Yaldad (B). DEGs were annotated using diverse databases (SwissProt, pfam, BLAST NR, eggNOG) and the species’ whole-genome sequence (GenBank BioProject no. PRJNA861856). Fold change values (FC_value_) of transplanted individuals are presented as negative numbers.

	Comparison	Up-Reg	DEG ID	FC_value_	Database	Database ID	Description
**(A)**	TCo vs. ACo	ACo	MCH015136.1	137.36	-	-	-
			MCH016307.1	39.17	eggNOG	10224.XP_006822956.1	Coagulation factor C-terminal domain
			MCH029407.1	30.40	-	-	-
			MCH014593.1	22.93	BLAST NR	XP_011447428.1	Complement C1q-like protein 4
			MCH027699.1	17.12	eggNOG	7955.ENSDARP00124702	Zinc ion binding
			MCH025331.1	14.51	BLAST NR	XP_022325980.1	NF-kappa-B-repressing factor-like
			MCH013756.1	12.28	pfam	PF00643.23	B-box zinc finger
			MCH027090.1	12.11	eggNOG	7739.XP_002593795.1	Ribonuclease H protein
			MCH009101.1	9.61	BLAST NR	AAQ83892.1	Interferon gamma-inducible protein 30
			MCH029447.1	9.17	SwissProt	K1PVI1_CRAGI	Iporin
		TCo	MCH017771.1	−108.62	SwissProt	A0A210QJ59_MIZYE	Solute carrier family 46 member 3
			MCH033811.1	−96.51	BLAST NR	OPL20729.1	Hypothetical protein AM593_09521
			MCH014167.1	−43.74	pfam	PF00160.20	Cyclophilin peptidyl-prolyl isomerase
			MCH019531.1	−43.29	SwissProt	A0A210Q3C1_MIZYE	Cyclic GMP-AMP synthase
			MCH008293.1	−34.14	eggNOG	59894.ENSFALP014062	Protein tyrosine phosphatase receptor
			MCH029089.1	−28.80	BLAST NR	OWF44506.1	GTPase IMAP family member 7
			MCH022781.1	−28.78	-	-	-
			MCH034174.1	−28.71	BLAST NR	OPL33663.1	Hypothetical protein AM593_02374
			MCH014606.1	−24.68	BLAST NR	XP_013385709.1	Periodic tryptophan protein 2 homolog
			MCH023306.1	−22.61	BLAST NR	XP_011445537.1	Late secretory pathway protein
**(B)**	TYa vs. AYa	AYa	MCH023206.1	198.83	pfam	PF04505.11	Interferon-induced transmembrane
			MCH018071.1	157.70	pfam	PF00098.22	Zinc knuckle
			MCH029089.1	100.05	BLAST NR	OWF44506.1	GTPase IMAP family member 7
			MCH033168.1	76.37	eggNOG	209285.XP_6694433.1	Cell wall integrity and stress response
			MCH002581.1	38.25	-	-	-
			MCH011954.1	22.44	SwissProt	A0A210Q5N4_MIZYE	PR domain zinc finger protein 2
			MCH034109.1	16.39	BLAST NR	OPL33917.1	Hypothetical protein AM593_04500
			MCH020918.1	15.13	pfam	PF12349.7	Sterol-sensing cleavage activation
			MCH028910.1	14.26	-	-	-
			MCH024145.1	11.40	SwissProt	A0A210QZX4_MIZYE	Beta-hexosaminidase
		TYa	MCH006622.1	−48.58	SwissProt	K1QES1_CRAGI	Peroxisomal NADH pyrophosphatase
			MCH025739.1	−24.57	BLAST NR	EKC32884.1	Aquaporin-2
			MCH012314.1	−24.02	BLAST NR	OPL32994.1	Hypothetical protein AM593_08485
			MCH008238.1	−23.74	BLAST NR	XP_022294261.1	Protocadherin Fat 4-like
			MCH029124.1	−23.07	-	-	-
			MCH012803.1	−20.50	pfam	PF00023.29	Ankyrin repeat
			MCH022944.1	−19.55	BLAST NR	OPL33555.1	Hypothetical protein AM593_04533
			MCH015808.1	−18.05	eggNOG	10224.XP_002740032.1	NACHT domain
			MCH020824.1	−17.64	eggNOG	10160.XP_004627003.1	Regulation of T-helper differentiation
			MCH032590.1	−15.26	pfam	PF00169.28	PH domain

**Table 4 ijms-26-00931-t004:** KEGG categorization of DEGs according to the local vs. foreign (δ_LF_) criterion for Cochamó (A) and Yaldad (B). The table lists up-regulated (Up-Reg) DEGs in each comparison, their associated KEGG term IDs, FDR *p*_value_, and a general description of the involved metabolic pathway.

	Comparison	Up-Reg	DEG ID	KEGG ID	FDR *p*_value_	Description
**(A)**	TYa vs. ACo	ACo	-	-	-	-
		TYa	MCH026485.1; MCH003699.1	crg00340	2.63 × 10^−3^	Histidine metabolism
			MCH026485.1; MCH003699.1	crg00380	4.21 × 10^−3^	Tryptophan metabolism
			MCH026485.1; MCH003699.1	crg00330	4.39 × 10^−3^	Arginine and proline metabolism
			MCH015372.1	crg03450	3.06 × 10^−2^	Non-homologous end joining
			MCH013290.1	crg00500	3.66 × 10^−2^	Starch and sucrose metabolism
			MCH026485.1; MCH003699.1; MCH002453.1; MCH013290.1	crg01100	3.66 × 10^−2^	Metabolic pathways
			MCH015372.1	crg03440	3.66 × 10^−2^	Homologous recombination
			MCH002453.1	crg00601	3.66 × 10^−2^	Glycosphingolipid biosynthesis
**(B)**	TCo vs. AYa	AYa	MCH015895.1	crg03060	9.86 × 10^−2^	Protein export
			MCH011254.1	crg04512	9.86 × 10^−2^	ECM–receptor interaction
			MCH011254.1	crg04145	1.41 × 10^−1^	Phagosome
			MCH000888.1	crg04144	1.54 × 10^−1^	Endocytosis
		TCo	MCH032590.1	crg00592	1.39 × 10^−1^	alpha-Linolenic acid metabolism
			MCH032590.1	crg00591	1.39 × 10^−1^	Linoleic acid metabolism
			MCH032166.1	crg00340	1.39 × 10^−1^	Histidine metabolism
			MCH032590.1	crg00565	1.39 × 10^−1^	Ether lipid metabolism
			MCH032166.1	crg00380	1.39 × 10^−1^	Tryptophan metabolism
			MCH032590.1	crg00590	1.39 × 10^−1^	Arachidonic acid metabolism
			MCH032166.1	crg00330	1.41 × 10^−1^	Arginine and proline metabolism
			MCH032590.1	crg00564	1.41 × 10^−1^	Glycerophospholipid metabolism
			MCH000692.1	crg04141	1.95 × 10^−1^	Protein processing in ER
			MCH007384.1	crg03040	1.95 × 10^−1^	Spliceosome
			MCH032166.1; MCH032590.1	crg01100	5.86 × 10^−1^	Metabolic pathways

**Table 5 ijms-26-00931-t005:** KEGG categorization of DEGs according to the *home* vs. *away* (δ_HA_) criterion for Cochamó (A) and Yaldad (B). The table lists up-regulated (Up-Reg) DEGs in each comparison, their associated KEGG term IDs, FDR *p*_value_, and a general description of the involved metabolic pathway.

	Comparison	Up-Reg	DEG ID	KEGG ID	FDR *p*_value_	Description
**(A)**	TCo vs. ACo	ACo	-	-	-	-
		TCo	MCH029783.1; MCH029782.1	crg00750	4.31 × 10^−4^	Vitamin B6 metabolism
			MCH029783.1; MCH029782.1	crg00260	7.35 × 10^−3^	Gly, Ser, and Thre metabolism
			MCH029783.1; MCH029782.1	crg00270	7.35 × 10^−3^	Cysteine Methionine metabolism
			MCH029783.1; MCH029782.1	crg01230	1.07 × 10^−2^	Biosynthesis of amino acids
			MCH029783.1; MCH029782.1	crg01200	2.15 × 10^−2^	Carbon metabolism
			MCH003699.1	crg00340	6.91 × 10^−2^	Histidine metabolism
			MCH003699.1	crg00380	9.72 × 10^−2^	Tryptophan metabolism
			MCH002447.1	crg00590	9.72 × 10^−2^	Arachidonic acid metabolism
			MCH003699.1	crg00330	1.03 × 10^−1^	Arginine and proline metabolism
			MCH002447.1; MCH029782.1; MCH003699.1; MCH029783.1	crg01100	1.17 × 10^−1^	Metabolic pathways
			MCH017106.1	crg04144	2.06 × 10^−1^	Endocytosis
**(B)**	TYa vs. AYa	AYa	MCH029782.1	crg00750	3.67 × 10^−2^	Vitamin B6 metabolism
			MCH015895.1	crg03060	6.96 × 10^−2^	Protein export
			MCH029782.1	crg00260	6.96 × 10^−2^	Glycine, Ser, and Thre metabolism
			MCH029782.1	crg00270	6.96 × 10^−2^	Cysteine Methionine metabolism
			MCH029782.1	crg01230	7.77 × 10^−2^	Biosynthesis of amino acids
			MCH029782.1	crg01200	1.03 × 10^−1^	Carbon metabolism
			MCH029782.1	crg01100	6.87 × 10^−1^	Metabolic pathways
		TYa	MCH032590.1	crg00592	7.36 × 10^−2^	alpha-Linolenic acid metabolism
			MCH032590.1	crg00591	7.36 × 10^−2^	Linoleic acid metabolism
			MCH032590.1	crg00565	7.36 × 10^−2^	Ether lipid metabolism
			MCH032590.1	crg00590	8.15 × 10^−2^	Arachidonic acid metabolism
			MCH032590.1	crg00564	8.90 × 10^−2^	Glycerophospholipid metabolism
			MCH012803.1	crg04142	1.51 × 10^−1^	Lysosome
			MCH032590.1	crg01100	7.01 × 10^−1^	Metabolic pathways

**Table 6 ijms-26-00931-t006:** Annotations of neighbor genes of outlier SNPs identified across different chromosomes. The table includes the outlier SNP ID, neighbor gene, associated databases and their IDs, and gene descriptions, providing insights into potential functional implications.

Chromosome	Outlier SNP ID	Neighbor Gene ID	Database	Database ID	Description
Chr 1	2352_34	MCH000058.1	BLAST NR	XP_021343742.1	uncharacterized protein C19orf44-like
Chr 1	6660_22	MCH000252.1	Swissprot	A0A210R0A2	protein phosphatase 1 regulatory subunit 42
Chr 1	6660_22	MCH000253.1	BLAST NR	XP_022314097.1	zinc finger SWIM domain-containing protein
Chr 1	7164_20	MCH000584.1	Pfam	PF13650.5	aspartyl protease
Chr 1	2467_50	MCH000860.1	BLAST NR	OWF55543.1	hypothetical protein KP79_PYT08876
Chr 1	2467_50	MCH000861.1	Swissprot	A0A1S3JHP6	probable rRNA-processing protein EBP2
Chr 1	13_74	MCH002388.1	-	-	-
Chr 1	13_74	MCH002389.1	-	-	-
Chr 1	2754_69	MCH002393.1	-	-	-
Chr 3	4790_45	MCH018322.1	-	-	-
Chr 3	4790_45	MCH018323.1	-	-	-
Chr 3	4790_45	MCH018324.1	-	-	-
Chr 3	9239_55; 6678_42	MCH018404.1	eggNOG	8010.XP_010901236.1	acid-sensing proton-gated ion channel
Chr 3	9239_55; 6678_42	MCH018405.1	eggNOG	10224.XP_006826007.1	zinc ion binding
Chr 4	836_12	MCH019553.1	eggNOG	7739.XP_002593026.1	centromere complex assembly
Chr 4	616_22	MCH019761.1	-	-	-
Chr 4	7299_61	MCH019944.1	BLAST NR	XP_022330633.1	ninein-like protein
Chr 4	8684_32	MCH019979.1	BLAST NR	OWF51276.1	neuronal acetylcholine receptor subunit a-3
Chr 4	8684_32	MCH019980.1	-	-	-
Chr 4	4041_7	MCH020207.1	BLAST NR	XP_021356125.1	transcription factor Sox-14-like
Chr 6	7434_34	MCH025417.1	Pfam	PF05721.12	phytanoyl-CoA dioxygenase
Chr 6	3097_32	MCH026223.1	eggNOG	28377.ENSACAP01319	sphingosine N-acyltransferase activity
Chr 6	1203_57	MCH026423.1	BLAST NR	AGU13048.1	myostatin
Chr 6	1870_40	MCH026543.1	Swissprot	K1QMC6	uncharacterized protein
Chr 6	1870_40	MCH026544.1	Swissprot	K1QGI1	uncharacterized protein
Chr 7	9179_59	MCH028892.1	-	-	-
Chr 7	9179_59	MCH028893.1	BLAST NR	XP_005093289.1	histidine triad nucleotide-binding protein
Chr 8	7678_48	MCH031237.1	eggNOG	10224.XP_002734847.1	B-cell translocation gene
Chr 8	6819_55	MCH031669.1	eggNOG	7739.XP_002597116.1	Maelstrom spermatogenic transposon silencer
Chr 8	6819_55	MCH031670.1	BLAST NR	XP_022327614.1	histone H4 transcription factor-like
Chr 8	6819_55	MCH031671.1	Swissprot	K1PE14	Pogo transposable element with KRAB domain
Chr 9	1292_19	MCH033323.1	eggNOG	6500.XP_005092544.1	homeobox protein unc-4 homolog
Chr 9	7598_24	MCH034354.1	Pfam	PF03732.16	retrotransposon gag protein
Chr 10	160_49	MCH003892.1	Pfam	PF00023.29	ankyrin repeat
Chr 10	2547_39	MCH004021.1	Swissprot	A0A210PPH0	protein mab-21-like 3
Chr 10	2547_39	MCH004022.1	BLAST NR	XP_021378478.1	protein mab-21-like 3
Chr 10	6715_15	MCH004774.1	Pfam	PF15433.5	mitochondrial 28S ribosomal protein S31
Chr 10	6715_15	MCH004775.1	BLAST NR	XP_021362947.1	osteopetrosis-associated transmembrane
Chr 11	3041_66	MCH006832.1	Pfam	PF03281.13	mab-21 protein
Chr 11	3041_66	MCH006833.1	Pfam	PF00643.23	B-box zinc finger
Chr 11	3041_66	MCH006834.1	eggNOG	10224.XP_002733914.1	RAB28, member RAS oncogene family
Chr 13	2134_59	MCH011299.1	Pfam	PF00022.18	actin
Chr 13	2134_59	MCH011300.1	BLAST NR	XP_021366265.1	dnaJ homolog subfamily C member 13-like
Chr 14	3073_57	MCH012310.1	Pfam	PF00654.19	voltage-gated chloride channel
Chr 14	3422_30	MCH012728.1	BLAST NR	EKC34371.1	Protein jagged-2

## Data Availability

All data generated and analyzed in this study, including Appendix A, are publicly available. The RNA-Seq raw reads are available as SRA runs in GenBank under the Bio Project accession number PRJNA630273.

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
