# Peer review of "Decoding Local Adaptation in the Exploited Native Marine Mussel *Mytilus chilensis*: Genomic Evidence from a Reciprocal Transplant Experiment"

_ijms, 2025, doi:10.3390/ijms26030931_

Round 1

Reviewer 1 Report

Comments and Suggestions for Authors

In general, it is a well done study. However, detailed results present rather obvious picture. Please see my comments and corrections in the attached file.It is obvious that the specimens transferred to another locality/habitat would be not adapted, but what about the stress caused by their transfer?

As concerns the interpretation of the results, and discussion, in my opinion the main problem concerns too simplicised or somewhat arbitrary statements. All the observed genotypic differences are interpreted as results of selection, which is too simplicized. What about the stochastic factors? Population history? Founder effect, bottlenecks, genetic drift?

Reviewer 2 Report

Comments and Suggestions for Authors

This is a very interesting and carefully conducted experimental study that shade light to the local adaptation of the edible and intensively harvested and cultured mussel Mytilus chilensis, by combining environmental biometrical and genetic data. Though it a lengthy MS, it is well written and of interest to a broad scientific audience as the presented results are comprehensively discussed under a main biological hypothesis. Congratulations! 

- The research shows the environmental effect on growth and adaptive gene expression in mussles by conducting a translocation experiment and thoroughly  analyse genomics,
- It is original research combinic genomic, biometrical, and environmental data that falls within the scope of the journal
- It clearly shows different gene expression under species adaptiveness
- Conclusion is informative addressing the tested hypotheses
- The references are appropriate
- I have no comment on tables/figures
